# Lower-Limb Amputation in Children and Adolescents—A Rare Encounter with Unique and Special Challenges

**DOI:** 10.3390/children9071004

**Published:** 2022-07-04

**Authors:** Axel Horsch, Svenja Gleichauf, Burkhard Lehner, Maher Ghandour, Julian Koch, Merkur Alimusaj, Tobias Renkawitz, Cornelia Putz

**Affiliations:** Department of Orthopaedics, Heidelberg University Hospital, 69118 Heidelberg, Germany; svenja.gleichauf@med.uni-heidelberg.de (S.G.); burkhard.lehner@med.uni-heidelberg.de (B.L.); mghandourmd@gmail.com (M.G.); julian.koch@med.uni-heidelberg.de (J.K.); merkur.alimusaj@med.uni-heidelberg.de (M.A.); tobias.renkawitz@med.uni-heidelberg.de (T.R.); cornelia.putz@med.uni-heidelberg.de (C.P.)

**Keywords:** amputation, lower extremity, complications, children, adolescents

## Abstract

Background/Aim: The pattern of lower-limb amputation, indications, complications, and revision in pediatric cases differs globally. Therefore, we conducted this study to describe the patterns of lower-limb amputation at our institution. Methods: During a set period between 2010 and 2020, adolescent patients undergoing lower-limb amputation within the orthopedic department of Heidelberg University Hospital were retrospectively collected and analyzed. The retrieved dataset included two parts: data on lower-limb amputations and data on subsequent complications and revision surgeries at the same time. Besides patients’ general information (age, gender), the dataset included data regarding amputation patterns (number, indications, and level of amputation, complications, and revision surgeries and their indications). Results: Twenty-two patients undergoing lower-limb amputation were examined, of which the majority were males (63.6%) with a mean age of 12 (5.1) years. Tumor was the most common indication for amputation (72.7%), and transfemoral amputation was the most frequent level (68.2%). Complications occurred in 10 patients, mostly due to stump impalement or bony overgrowth. Of all recorded patients requiring revision, nine were regarding bone and one case regarding soft tissue. Conclusions: Lower-limb amputation in adolescents is a rare encounter and it is commonly indicated due to bone tumors. The thigh is the most common level of amputation. Postoperative complications are frequent, mainly secondary to bony overgrowth, and often require revision surgery.

## 1. Introduction

Amputations of major limbs, especially in pediatric populations, are a rare encounter. In patients where amputation is necessary, the subsequent emotional and psychological impact is often detrimental to both the family and the affected child [1]. These major amputations of either the lower extremity or upper extremity lead to a substantial disfigurement of the amputated limb, carrying with it a fairly increased risk for postoperative morbidity and mortality, and especially children who suffer from limb amputation often perceive themselves as being incomplete [2]. In this context, creating a stump that is considered fit for a function-restoring prosthesis is essential for orthopedic surgeons.

The decision to amputate a limb follows at least one of several indications, including severe trauma, infection, tumors, congenital anomalies, and vascular abnormalities [1]. Evidence suggests that the reasons for amputations and their frequencies differ widely between countries. For instance, peripheral vascular affection is perceived as the leading cause of amputation in developed countries, while trauma, infection, and malignant tumors are reported as leading causes for major limb amputation in developing countries [3,4].

Although lower-limb amputations in pediatrics is rare, they are often associated with several complications, mostly being bone overgrowth regarding the amputated end of the limb and leading to permanent internal penetration of soft tissue under load, making prosthetic use impossible [5]. Another observed complication is the formation of a sharp-ended spur that potentially penetrates overlying soft tissue due to continuous growth, particularly in transfemoral (TF) and transtibial (TT) amputation. Addressing this, revision surgery may be considered, particularly to correct frequent growth abnormalities [6].

Due to the wide discrepancy in characteristics and outcome of pediatric patients undergoing major limb amputation within different countries, we conducted this research to study the pattern of lower-limb amputation in children and adolescents (<18 years of age), their indications, amputation levels, complications, and subsequent revision surgeries following amputation at our institution.

## 2. Materials and Methods

### 2.1. Study Population and Design

This research is based on a retrospective analysis of patient records containing lower-extremity amputations at the Orthopedic University Hospital Heidelberg (Figure 1). This retrospective analysis was based on two main datasets: (1) data of patients who underwent a lower-extremity amputation during the period from 2010 to February 2020 and (2) data on complications and revisions performed after amputation during the same period.

To create the dataset, all surgical procedures with a corresponding code were selected from the internal procedure list. Then, a review of the collected data using the ICD-10 and the coding of the surgical procedures to determine complications and performed revisions was conducted. Retrieved data contained baseline characteristics of patients (age, gender) as well as clinical information (level and indication of amputation, complications, and revisions).

### 2.2. Inclusion and Exclusion Criteria for Amputations

Inclusion criteria were set to patients containing following variables: lower-extremity amputations, any sex, and less than 18 years of age, at time of amputation surgery. For dually listed patients, we considered whether the amputation was performed on the same extremity, thus changing only the level of amputation or whether amputations were performed on different extremities. In case of two separately conducted amputations of the same limb, scoring took place as one patient; otherwise, this patient counted as two cases.

Patients that underwent an amputation other than lower extremity, as well as patients undergoing nonprimary amputation surgery, were excluded from this study. Therefore, the patient collective contained 23 patients. Reviewing inclusion criteria, 22 patients were thereafter relevant for further analysis.

### 2.3. Inclusion and Exclusion Criteria for Complications

To determine whether complications occurred because of initially conducted amputations or independently, an examination was performed within the patient cohort undergoing amputation of the lower extremity. If this was not the case, the patient would be excluded. If multiple listing of patients was detected, it was checked whether complications were co- or independent. If a continuing complication was detected, it counted as one patient and was therefore flagged with “multiple necessary revisions”. Independent complications were counted individually.

The patient population included a total of 10 patients, all of whom met the inclusion criteria and were therefore relevant for analysis.

### 2.4. Study Variables

Age, sex, cause of amputation, level of amputation, complications, and revisions performed were used as observation characteristics. A differentiated analysis of the above-mentioned characteristics was then considered.

The causes of amputation were divided into tumor, infection, dysmelia, and circulatory disorders. The amputation level was divided into TT amputation, TF amputation, hemipelvectomy (HP), and hip-disarticulation (HX). With regard to complications, only residual limb complications occurred in children and adolescents. Revisions were subdivided into soft-tissue and bony.

### 2.5. Statistical Analysis

Statistical analysis of the collected data was performed using Microsoft Excel 365 (Microsoft 2022, Redmond, Washington, DC, USA), and i.s.h.med (SAP Walldorf, Germany) to add missing information. Descriptive data were presented as means and standard deviations (SD) for continuous variables, while categorical and dichotomous variables were presented as numbers and frequencies.

## 3. Results

### 3.1. Demographic and Clinical Characteristics

The baseline characteristics of the retrieved records are summarized in Table 1. A total of 22 children/adolescents undergoing lower-extremity amputation were included in this analysis. The mean age was 12 (SD = 5.1) years, and 63.6% (14/22) of patients were males. The mean follow-up was 4.3 years (SD = 3.4).

Amputation was conducted in all 22 patients with tumor being the most common reason for amputation, accounting for 16 patients (72.7%), followed by dysmelia (13.6%) and circulation (9.1%), respectively. Of patients with tumor as indication for amputation, osteosarcoma was the most commonly reported cause (15/16 patients; 93.8%), while Ewing’s sarcoma was reported in one case. TF amputation was most common accounting for 68.2% of patients, followed by HX (13.6%), TT amputation (9.1%), and HP (9.1%), respectively.

### 3.2. Complications following Amputations

Complications occurred in 10 (45.45%) out of 22 amputated patients, all of which were residual limb complications. Based on an intact growth plate, which is present in transmetaphyseal amputations (at femoral and tibial levels) (Figure 2), impaling of the residual limb occurred. This happens mainly due to continuous longitudinal growth of the bone leading to excessive stress on soft tissue. As a result, 9 of 17 patients (52.9%) experienced stump impalement (Figure 3).

In four of the nine recorded patients with limb impalement as a complication, a shorting of the bone and a remolding of the distal end of the stump (deperiosteostomy) was performed during amputation. This procedure was not used in the remaining five patients.

### 3.3. Revisions Secondary to Complications

Revision surgeries following amputation were performed in 10 patients that experienced postoperative complications, 9 of which received bony revisions accounting for 90% of all revisions, while 1 case had soft tissue revision. A summary of bony revision procedures that were carried out is provided in Table 2. Stump shortening and stump revision was most common in our population, accounting for 33.3% of revisions.

## 4. Discussion

Major limb amputation has been perceived as a frequent surgical procedure attempted in many orthopedic, general, vascular, and trauma settings, mainly for life-saving purposes. That being said, amputation is commonly associated with a substantial impact on the affected individual’s social and psychological status [7,8]. The patterns of amputation indications, frequency, and levels, as well as subsequent complications and need for revision surgeries, are widely variable based on the age of patients, settings, and country in which they are carried out [1,6]. Therefore, we conducted this retrospective analysis of children and adolescent patients who underwent lower-limb amputation at our institution over a 10-year period to report patterns of amputation indications, complications, and revision surgery.

A total of 22 patients were included for descriptive analysis, following the exclusion of 1 case that did not match our eligibility criteria. All of them were children or adolescents with a mean age of 12 years. We observed a male predominance accounting for 63.6% of amputated patients. This observation stands in line with the recent literature [9,10,11,12,13]. Although, most of the available evidence provides data in adult patients, and it therefore seems that male predominance in amputation is a common observation across all age groups. However, this still warrants further investigation to determine the reasons behind this predominance, as the presented sample size is too small for a principal generalization.

Indications for limb amputation, particularly within pediatric population, vary greatly in different settings and countries. For instance, some reports indicate almost 80 to 90% of limb amputations in adults occur as a result of vascular abnormality in developed countries [14,15,16] while in developing countries, trauma and infections are perceived as the most prevalent indications for limb amputations [3,4]. It is assumed that in developed countries, pediatric amputations are mainly performed to treat tumors, while in developing countries, trauma and infections are the most common indications. In this study, tumors were the leading cause of amputation in children/adolescents, followed by dysmelia. This is somewhat consistent with the study by Chalya et al. [6], where vascular abnormalities were ranked third as a cause of lower-limb amputation; however, diabetic foot and trauma were ranked first and second, respectively. Of note, the authors included individuals of all age groups, while most of them had an age ranging between 41 and 50 years. Similar to the previous study, Jahmani et al. [5] reported diabetic foot (41.9%) and trauma (38.4%) as leading indications for amputation. The authors studied indications for lower-limb amputation in patients with (0-10) and (11-20) years of age, of which primary bone cancer and trauma were reported as the common indication for amputation within each of these age groups, respectively. The differences in indication for amputation between this study and recent literature could be related to this unique and young age group within the analyzed cohort.

Of note, in our population, the decision of which type of amputation was to be carried out was solely dependent on the findings of tumor extension, while putting into consideration an appropriate safety margin. In our study, TF amputation was the most frequent level of lower-limb amputation among children and adolescents. This is consistent with current literature, where above-knee amputation was reported as the most common level of amputation, both in the adult and pediatric population [10,17,18].

We observed many complications in our patients undergoing lower-limb amputation, accounting for 45.5% of the studied population. All observed complications were residual limb complications, with stump impalement or bony overgrowth being most frequent (nine patients). In four of nine patients, deperiosteostomy was performed to tackle this complication. Preservation of the length of limb in this population is of great importance, mainly because the lost distal growth plate accounts for approximately 75% of vertical femoral growth. In these patients, TF amputations lead to a very short stump leading to a complex biomechanical situation regarding prosthetic fitting. On the other hand, TT amputation, although preserving the proximal growth plate, often leads to bony overgrowth at the distal end, which subsequently may continue growing until growth is stationary within amputated individuals [1]. Consistent with our findings, bony overgrowth is reported as the most prevalent complication following lower-limb amputation in adolescents, with rates ranging from 4 to 50% [19,20,21,22,23,24]. Importantly, age, location, indication for amputation, and level of amputations are perceived as attributable factors for such prevalence, whereas younger individuals are more likely to experience this complication [19,20,25]. This could be the reason for this complication being so frequent in our population, secondary to the young age of included individuals.

It is important to highlight an unfamiliar observation in one of our cases, where the fibula outgrew the tibia following amputation (Figure 4). Although the surgery was performed correctly and no epiphysiodesis was performed, this observation remains unclear.

More so, all amputated individuals in our study who experienced postoperative complications had undergone revision surgery (10 out of 10 patients). Little is reported regarding the value of revision surgery on lower-limb amputation, particularly in the pediatric population due to the scarcity of evidence in this particular field. In a recent analysis of 71 revision surgeries in patients who underwent lower-limb amputation, it was reported that soft-tissue pathology (31.0%) was the most common indication for revision surgery followed by infections (31.0%) and bone pathology (18.3%), respectively [26]. This differs from our observation, since in our series, nine patients underwent bony revision, whilst only one case underwent soft tissue revision surgery. This discrepancy could be related to indication for amputation in the first place, as in our study, tumor (mainly osteosarcoma) was the main indication for amputation, whereas trauma was most common for amputation in the previous study [26]. This highlights an important observation relating to the association between the primary indication for amputation and subsequent complications and revision surgeries. Nevertheless, larger and prospective studies are needed to further investigate this observation. That being said, it is of great importance to mention that this problem of bony impalement begins early in these patients, and due to the bone overgrowth, there is a significant stress on the surrounding soft tissues, leading to unequivocal growth and eventually penetration. This in turn results in a significant impact of affected patients’ life in terms of pain and activities of daily living which can also negatively impact the affected child’s mental health. In these circumstances, revision surgery is considered a necessary function-improving intervention.

Surgical options for revision surgery in children and adolescents include stump shortening with the option of temporary epiphysiodesis of the proximal growth plates. However, the resulting reduced bone growth can complicate restoration in very short residual limbs and lead to instability in the overlying joint, making length preservation/gain advantageous for prosthetic restoration. Even an operation according to Ertl does not solve the problem of the bone growing further. The orthopedic technical restoration with traction liners and the prosthetic restoration with stretching/tensioning of the soft tissues (i.e., pin system) should be optimized accordingly to reduce revision interventions during growth.

Of note, infantile amputation is a rare occurrence and our study is the first to provide valuable insights into lower-limb amputation patterns and complications in children and adolescents in Germany over a 10-year period. Despite the high revision rate due to bony impalement, there is no alternative to amputation in terms of providing good functional outcome. That being said, our study has several limitations, the biggest of which being the limited sample size; however, because lower-limb amputation in pediatric populations is rare, we could analyze this population’s characteristics in a larger dataset, even though a retrospective design limits the conclusions that could be drawn from it, given the lack of important data outlining a better perspective on this patient population. Therefore, future research should include a larger, more widely representative dataset to further investigate, describe and analyze lower-limb amputation patterns in pediatric patients.

## 5. Conclusions

Lower-limb amputation in children and adolescents is a rare encounter, which is more predominant in males and is commonly indicated due to bone tumors. TF amputation is the most common level of amputation. Postoperative complications are reported in nearly half the amputated population, besides bony overgrowth. Most of these complications indicate and result in revision surgery to correct for the occurring abnormality, of which stump growth is most common. Despite the high revision rate, it must be noted that amputation is without alternative in these cases. In addition, modern prosthesis fitting usually enables patients to have very good everyday function.

## Figures and Tables

**Figure 1 children-09-01004-f001:**
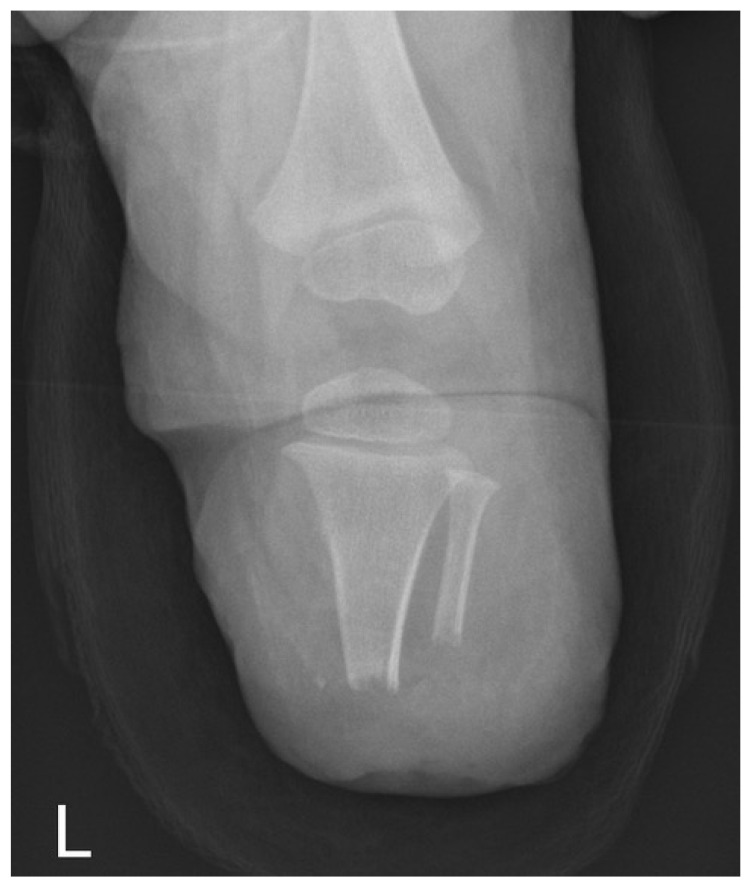
X-ray image of a patient included in our dataset.

**Figure 2 children-09-01004-f002:**
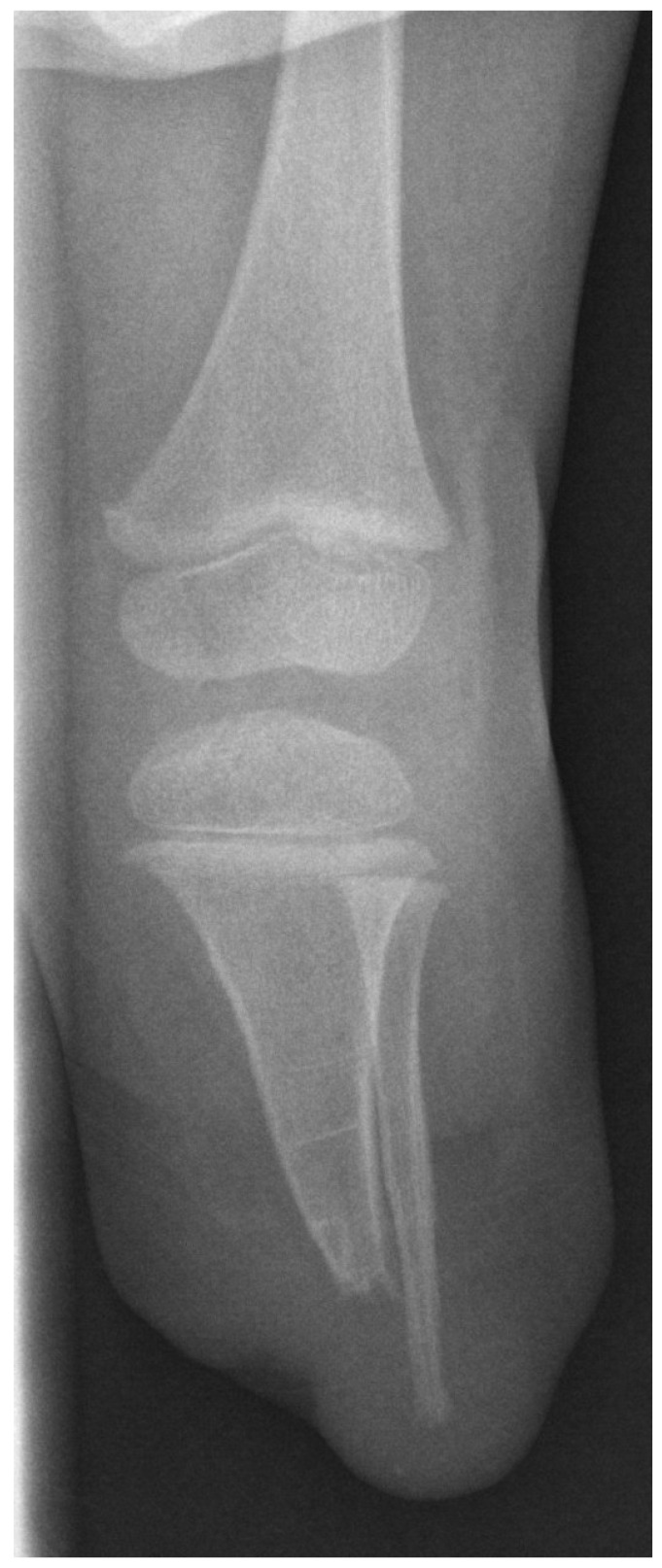
Postoperative X-ray of transtibial amputation.

**Figure 3 children-09-01004-f003:**
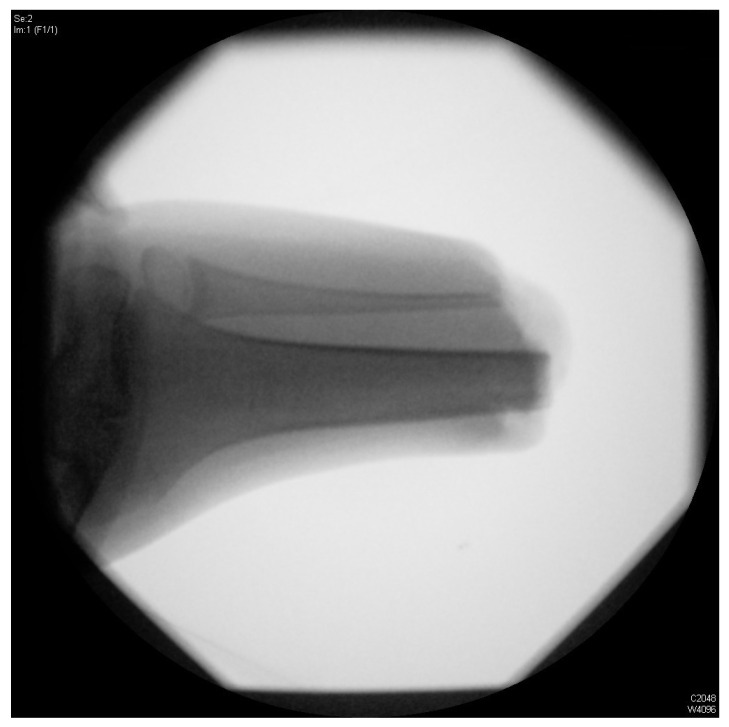
X-ray image of stump spike.

**Figure 4 children-09-01004-f004:**
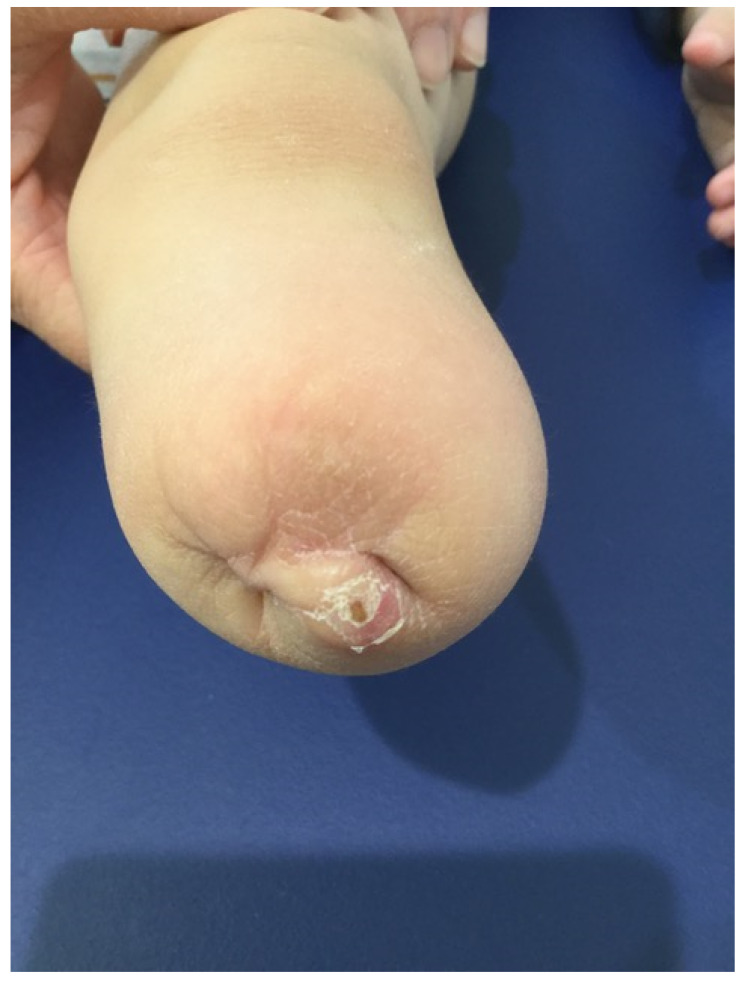
A clinical image showing an unfamiliar observation of the fibula outgrowing the tibia in an amputated case.

**Table 1 children-09-01004-t001:** Demographic and clinical characteristics of included children/adolescents with lower-extremity amputation (*n* = 22).

Variable	Subcategory	Total Population (*n* = 22)
Age (years); mean (SD)		
		12 (5.1)
Gender; *n* (%)		
	Male	14 (63.6%)
	Female	8 (36.4%)
Causes of Amputation; *n* (%)		
	Tumor	16 (72.7%)
	Infection	1 (4.5%)
	Dysmelia	3 (13.6%)
	Circulation	2 (9.1%)
	Total	22 (100%)
Level of Amputation; *n* (%)		
	Transtibial	2 (9.1%)
	Transfemoral	15 (68.2%)
	Hemipelvectomy	2 (9.1%)
	Hip-disarticulation	3 (13.6%)
	Total	22 (100%)
Complications; *n* (%)		
		10 (45.45%)
Revision Surgery due to complications; *n* (%)		
		10 (45.45%)

*n*: Number; SD: Standard Deviation.

**Table 2 children-09-01004-t002:** Summary of the bony revision procedures that were performed in complicated amputated patients.

Variables	Number (%)
Revision with stump shortening + reconstruction with bone cartilage cap plasty	1 (11.1%)
Stump revision with ablation of bone tips	1 (11.1%)
Stump shortening and stump revision	3 (33.3%)
Stump shortening with periosteoplasty	2 (22.2%)
Stump cap plasty with bone wedge from iliac crest	1 (11.1%)
Resection of exostosis and bony aspects	1 (11.1%)
Total	9 (100%)

## Data Availability

All the Data analyzed in this manuscript can be provided upon request by contacting the corresponding author.

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
