# Peer review of "Lower-Limb Amputation in Children and Adolescents—A Rare Encounter with Unique and Special Challenges"

_children, 2022, doi:10.3390/children9071004_

Round 1

Reviewer 1 Report

Thank you for submitting “Lower Limb Amputations in Children – A Rare Encounter with Unique and Special Challenges”. This is a retrospective review of lower-extremity amputations in children at a single center.

I recommend editing the methods for clarity. It reads as if only primary amputations that had complications were included in this paper, however, the results seem to suggest that this is not the case.

The paper specifically discusses “adolescents”; however, the inclusion was <18 years old and the range was from 6.9-17.1yrs. Length of follow-up is not mentioned in the paper and should be included

Most of the time, amputations in children are attempted through joint (Boyd, Syme, knee disarticulation). There were no through-joint amputations in your group other than hip disarticulations?

High revision rates with trans-tibial and Trans-femoral amputations in children is well known. What is added by the publication of this paper?

Author Response

We would like to thank the author for his/her great contribution to the quality of the current version of the manuscript. Attached is a point-by-point summary of our replies/edits. All edits in the manuscript can be tracked through the "Track changes" option in word.

1- I recommend editing the methods for clarity. It reads as if only primary amputations that had complications were included in this paper, however, the results seem to suggest that this is not the case.

Response: Thank you for highlighting this point. We clarify that the choice of words might haven’t been very well selected. However, we want to clarify that patients with lower limb amputations were included in our retrospective study regardless of the status of postoperative complications.

Meanwhile, such complications were then screened and reported in this dataset.

To further clarify, we have attached a new figure relating to the radiographic presentation of one of our cases at baseline (no accompanying complication). We have edited the first sentence of the Methods section as follows: “This research is based on a retrospective analysis of patient records containing lower extremity amputations at the Orthopedic University Hospital Heidelberg.“

2- The paper specifically discusses “adolescents”; however, the inclusion was <18 years old and the range was from 6.9-17.1yrs. Length of follow-up is not mentioned in the paper and should be included

Response: Thank you for your great advice and for bringing this to our attention. We have edited this part throughout the whole manuscript, including the title and keywords, to be [children and adolescents“ instead of [children alone].

As for the follow-up period, we have added it to the results section as follows: “The mean follow-up was 4.3 years (SD=3.4).“

3- Most of the time, amputations in children are attempted through joint (Boyd, Syme, knee disarticulation). There were no through-joint amputations in your group other than hip disarticulations?

Response: We appreciate you raising this point. We would like to highlight the fact that the corresponding amputations were made based on the diagnosis and the diagnostic MRIs before the operation. The majority of our population (75%) had tumors showing different extensions, where an appropriate safety margin should be considered to avoid recurrence. In our dysmelia cases, the transtibial stumps are more preferred knee exacerbations; however, in our cases, the knee was preserved due to the absence of these points and therefore, transtibial amputations were attempted. We have also added this sentence in the discussion section to further clarify this: “Of note, in our population, the decision of which type of amputation was be done was solely dependent on the findings of tumor extension, while putting into consideration an appropriate safety margin.“

4- High revision rates with trans-tibial and Trans-femoral amputations in children is well known. What is added by the publication of this paper?

Response: We appreciate your comment. However, our study adds novelty to the available body of evidence in these regards:

  • Our study is novel in the fact that it examined the largest collective of children and adolescents with lower extremity amputations. And, it also determined exactly how high the proportion of revision surgery and complications is, particularly in young patients with bone tumors.
  • Another thing is that surgically, except for stopping the growth by means of temp. epiphyseodesis and stump shortening, there is no alternative. The orthopedic care with traction liners and the prosthetic care with stretching/pulling of the soft tissues, e.g. pin system, should be optimized accordingly.

Reviewer 2 Report

There are a few questions and comments I have as well as suggestions for improvement of this manuscript.

1) Figures 1 shows an overlong fibula and a normal length of the tibia. During amputation, the fibula is always dissected shorter. Why has the fibula grown so much here and the tibia not? This should be better described as it is unusual for the fibula to overgrow the tibia. Was anything done in the epiphysis in this patient?

2) Figure 2 An additional clinical pictures showing the stump spike would make this phenomenon even clearer, especially if there are still visible soft tissue lesions.

3) The statements with literature reference no. 26 between lines 202 and 212 should be better presented with new literature references to be discussed again. Here a comparison of the results is made with a paper where the main causes of aputation are trauma. The fact that the complication rate for traumatic amputations is higher than for other causes is widely known and often cited. For this reason, a comparison should be made with studies where the causes are not mainly traumatic.

4) The authors should still describe possibilities or recommendations on how such complications can be prevented or avoided in principle.

Author Response

We would like to thank the author for his/her great contribution to the quality of the current version of the manuscript. Attached is a point-by-point summary of our replies/edits. All edits in the manuscript can be tracked through the "Track changes" option in word.

1) Figures 1 shows an overlong fibula and a normal length of the tibia. During amputation, the fibula is always dissected shorter. Why has the fibula grown so much here and the tibia not? This should be better described as it is unusual for the fibula to overgrow the tibia. Was anything done in the epiphysis in this patient?

Response: Thank you for highlighting this point. Unfortunately, in this particular patient, everything was done correctly and according to the protocol. No epiphysiodesis was performed. And, it remains unclear why the fibula has gained so much extra length.

We added this sentence in the discussion section: “It is important to highlight an unfamiliar observation in one of our cases, where the fibula outgrew the tibia following amputation (Figures 1). Although the surgery was done correctly and no epiphysiodesis was performed, this observation remains unclear.“

2) Figure 2 An additional clinical picture showing the stump spike would make this phenomenon even clearer, especially if there are still visible soft tissue lesions. 

Response: Thank you for your comment. Unfortunately, a clinical photograph of this exact case is unavailable, and we have added another photograph of another patient for your use and reference (please check the attached image). If you think that this photograph would be good for presentation in the manuscript, please let us know and we'll add it to the manuscript.

3) The statements with literature reference no. 26 between lines 202 and 212 should be better presented with new literature references to be discussed again. Here a comparison of the results is made with a paper where the main causes of amputation are trauma. The fact that the complication rate for traumatic amputations is higher than for other causes is widely known and often cited. For this reason, a comparison should be made with studies where the causes are not mainly traumatic.

Response: Thank you for your valuable comment and suggestion. We would like to point out that this is the first study with the largest collective of children and adolescents with a focus on tumor diseases who have received an amputation and there are no other comparable results in the literature in this regard. And, unfortunately, in the available body of evidence, there are not many studies similar to us in respect to design, sample size, and patient characteristics or indications for amputation.

4) The authors should still describe possibilities or recommendations on how such complications can be prevented or avoided in principle.

Response: Thank you for your valuable suggestions and help in improving the quality of our manuscript. We have added this sentence at the end of the discussion section right before the limitations part as follows: “Surgical options for revision surgery in children and adolescents include stump shortening with the option of temporary epiphysiodesis of the proximal growth plates. However, the resulting reduced bone growth can complicate restoration in very short residual limbs and lead to instability in the overlying joint, making length preservation/gain advantageous for prosthetic restoration. Even an operation according to Ertl does not solve the problem of the bone growing further. The orthopedic technical restoration with traction liners and the prosthetic restoration with stretching/tensioning of the soft tissues (i.e., Pin System) should be optimized accordingly to reduce revision interventions during growth.“

Round 2

Reviewer 2 Report

2) Figure 2 An additional clinical picture showing the stump spike would make this phenomenon even clearer, especially if there are still visible soft tissue lesions. 

Response of the Authors: Thank you for your comment. Unfortunately, a clinical photograph of this exact case is unavailable, and we have added another photograph of another patient for your use and reference (please check the attached image). If you think that this photograph would be good for presentation in the manuscript, please let us know and we'll add it to the manuscript.

My Response: the photo shows the problem and gives the paper more value. Please add the photo with short description.

Author Response

Dear reviewer, thank you so much for your valuable contribution and insightful comments. We added the figure (Fig 4) with a short description revealing this uncommon observation. Again, we appreciate your efforts in improving the quality of our paper.